# Root morphological and physiological characteristics in maize seedlings adapted to low iron stress

Wengjing Long[1], Qiang Li[2], Nianxin Wan[3], Dongju Feng[4], Fanlei Kong[4], Yong Zhou[2], Jichao Yuan[4]*

1 Rice and Sorghum Institute, National Sorghum Improvement Center Sichuan Branch, Sichuan Academy of Agricultural Sciences, Deyang, China, 2 Chongqing Key Laboratory of Economic Plant Biotechnology, Collaborative Innovation Center of Special Plant Industry in Chongqing, Institute of Special Plants, Chongqing University of Arts and Sciences, Yongchuan, China, 3 Luxian Management Committee of modern agricultural park, Luzhou, China, 4 Key Laboratory of Crop Ecophysiology and Farming System in Southwest China, Ministry of Agriculture, Sichuan Agricultural University, Chengdu, China

☯ These authors contributed equally to this work.
* yuanjichao5@163.com

**Data Availability Statement:** All relevant data are within the manuscript and its Supporting Information files.

## Abstract

Iron (Fe) deficiency is a common challenge in crop production. Screening and research of Fe-efficient cultivars could alleviate plant stress and increase crop yields in Fe-deficient soils. In the present study, we conducted two hydroponic culture experiments with a control (100 µmol/L $Fe^{3+}$-EDTA) and low Fe treatment (10 µmol/L $Fe^{3+}$-EDTA) to study the morphological and physiological mechanisms of response to low Fe stress in maize hybrids seedlings. In the first experiment, we investigated 32 major maize hybrids in Southwest China. We found that six of them, including Zhenghong 2 (ZH 2), were Fe-efficient. Fifteen other cultivars, such as Chuandan 418 (CD 418), were Fe-inefficient. In the second experiment, we investigated the Fe-efficient ZH 2 and Fe-inefficient CD 418 cultivars and found that low Fe stress resulted in significant decreases in root volume, root length, number of root tips, root surface area, and root dry weight, and increased root to shoot ratio, average root diameter, and Fe-dissolution ability per mass of roots in both maize cultivars. However, the increase in Fe-dissolution ability per mass of roots in ZH 2 was higher than that in CD 418, whereas for the other measurements, the low Fe stress-induced changes in ZH 2 were less pronounced than in CD 418. Therefore, under low Fe stress, the above-mentioned growth factors in ZH 2 were higher by 54.84%, 121.46%, 107.67%, 83.96%, 140.00%, and 18.16%, respectively, than those in CD 418. In addition, leaf area, chlorophyll content, net photosynthetic rate, soluble protein content, and Catalase (CAT) and Peroxidase (POD) activities in ZH 2 were higher by 274.95%, 113.95%, 223.60%, 56.04%, 17.01% and 21.13% than those in CD 418. Therefore, compared with the Fe-inefficient cultivar (CD 418), the Fe-efficient cultivar (ZH 2) had a more developed root system and greater Fe absorption capacity per mass of roots under low iron stress, promoted the efficient absorption of Fe, maintained a higher photosynthetic area and photosynthetic rate, thereby facilitating the accumulation of photosynthetic products. Moreover, higher soluble protein content and activities of CAT and

**Funding:** This study was financially supported by the National Key Research and Development Program for Grain High-yield Science and Technology Innovation Project of China (Grant No. 2016YFD0300307, 2017YFD0301704, 2018YFD0301206), and the Special Fund for Agro-scientific Research in the Public Interest of China (Grant No. 20150312705) to JY.

**Competing interests:** The authors have declared that no competing interests exist.

POD permitted high osmotic regulation and scavenging ability, which is an important physiological mechanism for ZH 2 adaptation to low Fe stress.

## Introduction

Maize (*Zea mays* L.) is the world's most produced food crop. Total annual maize production exceeds 1 billion tons and accounts for 41% of the world's total grain production [1–2]. Therefore, maize plays a significant role in global food security [3]. Iron (Fe) deficiency considerably restricts maize production, hence threatening food security. Fe is an essential element for plants and plays a vital role in the metabolism of matter and energy [4, 5]. It is the fourth most abundant element in the earth's crust, but due its tendency to form oxidized and hydroxide states under oxidizing, neutral, or alkaline soil conditions, plant-available ionic Fe concentrations in soil are very low. Plants often show symptoms of Fe deficiency, especially in arid and semi-arid calcareous soil areas [6, 7]. Sichuan Province is a big province of maize planting and consumption in China. More than 40% of dry land in the region has calcareous purple soil, and the content of available iron in soil is low, which seriously restricts the improvement of the maize yield [8, 9].

Fe deficiency is a critical challenge to plant growth because Fe is involved in several key chemical processes. For example, Fe is involved in the synthesis of chlorophyll, the vital pigment required for photosynthesis, the most important physiological process in plants. About 80% of Fe in plants is concentrated in chloroplasts [10, 11]. In addition to its importance to photosynthesis, Fe is involved in cell detoxification. Fe is part of the active center of the cell defense enzyme system for catalase and peroxidase [6]. Maize growth is significantly inhibited by Fe deficiency, which results in reduced chlorophyll and active Fe content [12, 13].

Under conditions of Fe deficiency, a series of physiological and biochemical changes occur to promote Fe absorption to maintain plant growth and development [12]. As a mechanism II plant, the most important metabolic process in maize for Fe absorption is the synthesis and secretion of the Fe carrier mugineic acid. When deficient in Fe, maize synthesizes and actively secretes mugineic acid, which chelates trivalent Fe in rhizosphere soil and transports Fe into root cells by the action of a specific transporter [10, 14]. However, there are significant differences in the response of different maize cultivars to mugineic acid secretion and low Fe stress [13]. Therefore, breeding and promotion of Fe-efficient maize cultivars is a potential strategy to alleviate low Fe stress and improve maize yield in Fe-deficient soils.

Absorption and utilization of Fe by plants is a complex process closely related to morphological and physiological characteristics which can vary between plant cultivars [15, 16]. For example, Gao et al. [17] studied the differences in leaf yellowness, chlorophyll content, and active Fe content of 16 peanut cultivars in calcareous soil and found that chlorophyll content in the early growth stage is highly related to yellowness, active Fe, and yield. Chen et al. [18] studied the differential response of 12 maize inbred lines to low iron stress, and found that the iron absorption efficiency of 12 inbred lines varied from 5–40%, among which, the iron absorption efficiency of DE3 was the highest, while that of B77 was the lowest.

Whereas previous studies on efficient absorption and utilization of Fe mainly focused on the mechanism I plants, such as Arabidopsis and apple [10], there are few reports on mechanism II plants such as maize [13]. In addition, the morphological and physiological mechanisms of maize adaptation to low Fe stress remains unclear. It is not known if there are significant differences in root morphology and physiological characteristics in maize seedlings

with different iron efficiencies adapted to low iron stress. Therefore, in the present study, 32 maize hybrids accounting for over 90% of the total planting area of maize in Southwest China were collected and studied. Two hydroponic culture experiments were conducted to assess the difference in Fe absorption and utilization between the hybrids, and to explore the root morphological and physiological mechanisms of maize seedling adaptation to low Fe stress. Overall, our study provides novel insights into theoretical basis for efficient absorption and utilization of Fe nutrition by maize.

## Materials and methods

### Experimental conditions

Two hydroponic culture experiments were conducted in Wenjiang, Southwest China. The first experiment was conducted in 2015 with 32 maize hybrids that are commonly cultivated in Southwest China (S1 Table). The second experiment was conducted in 2016 with an Fe-efficient cultivar, ZH 2, and an Fe-inefficient cultivar, CD 418. The differences in the main indices of the two cultivars in the first experiment were significantly different under low iron stress.

### Experimental design

Both experiments were set up in a randomized complete block design. Maize seedlings were assigned to either a control or low Fe treatment, with three replicates having 20 plants in each replicate in the first experiment, and 60 plants in each replicate in the second experiment. To grow seedlings for study, maize seeds were sterilized with 10% (v/v) hydrogen peroxide for 30 min, washed six times with distilled water, and immersed in distilled water for 12 h. The seeds were then germinated in a light incubator at 60% relative humidity and a light: dark cycle of 14:10 h (26:22˚C). Once maize seedlings had two expanded leaves, seedlings with consistent growth were selected, their endosperms were removed, and they were transferred to black plastic pots containing 10 L of modified Hoagland nutrient solution (100 μmol/L $Fe^{3+}$-EDTA). The basic nutrient solution contained 2.0 mM $Ca(NO_3)_2$, 0.75 mM $K_2SO_4$, 0.1 mM KCl, 0.25 mM $KH_2PO_4$, 0.65 mM $MgSO_4$, 1.0 μM $MnSO_4$, 1.0 μM $ZnSO_4$, 0.1 μM $CuSO_4$, and 0.005 μM $(NH_4)_6Mo_7O_{24}$. Seedlings in the low Fe treatment received only 10 μmol/L $Fe^{3+}$-EDTA once they grew three expanded leaves, whereas control seedlings used the modified Hoagland nutrient solution (100 μmol/L $Fe^{3+}$-EDTA). The seedlings were cultured in a growth chamber with a light: dark cycle of 14:10 h (28:22˚C). The nutrient solutions were changed every four days. The pH of the nutrient solution was adjusted to 5.8 with NaOH and was ventilated using a pump.

### Sampling and measurements

For the first experiment, plant height, stem diameter, the number of visible leaves, leaf area, chlorophyll content, root morphology, dry matter, and active Fe content were measured on 10 plants × 3 replicates per treatment, seven days after assignment to low Fe or control treatments. For each measurement, the average value was calculated from the 10 plants within each replicate and used for analysis. Definitions for each measurement were as follows: plant height —the length of the coleoptile node to the highest leaf tip measured with a ruler; stem diameter —the maximum diameter at the base of seedlings, 1 cm away from the root-measured with a Vernier caliper; number of visible leaves—the number of all visible leaves including the heart leaf; leaf area, assessed with the length-width coefficient method (length × width × 0.75); chlorophyll content—the relative content of chlorophyll in the youngest fully expanded leaf, determined with a SPAD chlorophyll meter instrument (SPAD-502Plus); root morphology—the

root surface area, root volume, root length, average root diameter, and number of root tips analyzed with an Epson Expression 1000Xl (Seiko Epson Corp., Suwa, Japan) scanner and WinRHIZO (Regent Instruments Inc., Quebec, Canada); dry matter accumulation—the dry weight and root to shoot ratios of maize seedlings divided into shoots and roots, killed at 105˚C for 30 min, dried at 80˚C to constant weight, and weighed; active Fe content [18]—fresh leaf samples were chopped and extracted with 1 mol/L HCl at a ratio of 1:10, and after filtering, the content of Fe in the extraction solution was measured by an atomic absorption spectrophotometer; and Fe content [19]—as determined by atomic absorption.

For the second experiment, plant height, stem diameter, dry matter, and Fe content were measured 14 and 28 days after treatment. In addition, root morphology, root activity, mugineic acid content, leaf area, chlorophyll content, photosynthetic rate, and other physiological indices were measured 14 days after treatment. All measurements were taken on 3 replicates (10 plants per replicate). Methods and sample numbers for plant height, stem diameter, dry matter, Fe content, root morphology, leaf area, and chlorophyll content were the same as those in the first experiment. Methods for other measurements were as follows: root activity [20], determined by the Triphenyl tetrazolium chloride (TTC) reduction method; mugineic acid content (Fe-dissolution ability) [21]—determined by atomic absorption; photosynthetic rate—measured using a Li-6400 photosynthesis instrument with 6 leaves measured per replicate; catalase (CAT) activity [22]—determined by potassium permanganate titration; peroxidase (POD) activity [2]—measured by the guaiacol method; soluble protein content [23]—determined by Coomassie brilliant blue G250 method.

## Calculations [23, 24]

$$\text{Leaf area} = \text{Length} \times \text{width} \times 0.75 \qquad (1)$$

$$\text{Fe accumulation} = \text{Fe content} \times \text{dry weight} \qquad (2)$$

$$\text{Low Fe tolerance coefficient} = \frac{\text{value under low Fe treatment}}{\text{value under control treatment}} \qquad (3)$$

$$\text{Fe physiological efficiency} = \frac{\text{dry weight}}{\text{Fe accumulation}} \qquad (4)$$

## Statistical analysis

The experimental data were sorted in MS Excel 2010 (Microsoft Corp., Redmond, WA, US) and were analyzed using IBM SPSS Statistics 21.0 (IBM Corp., Armonk, NY, US). Means were compared using the least significant difference at the 0.05 significance level to determine the differences between the means of each treatment. Graph-Pad Prism v. 5.0 (GraphPad Software Inc., La Jolla, CA, USA) was used for mapping.

## Results

### Cultivar screening

Plant height, stem diameter, number of visible leaves, leaf area, root dry weight, shoot dry weight, dry weight per plant, relative chlorophyll content, root volume, root length, primary root length, number of root tips, root surface area, Fe content, and Fe accumulation were significantly lower under the low Fe treatment compared with the control treatment (low Fe tolerance coefficient < 1), whereas leaf active Fe content, root to shoot ratio, and root diameter

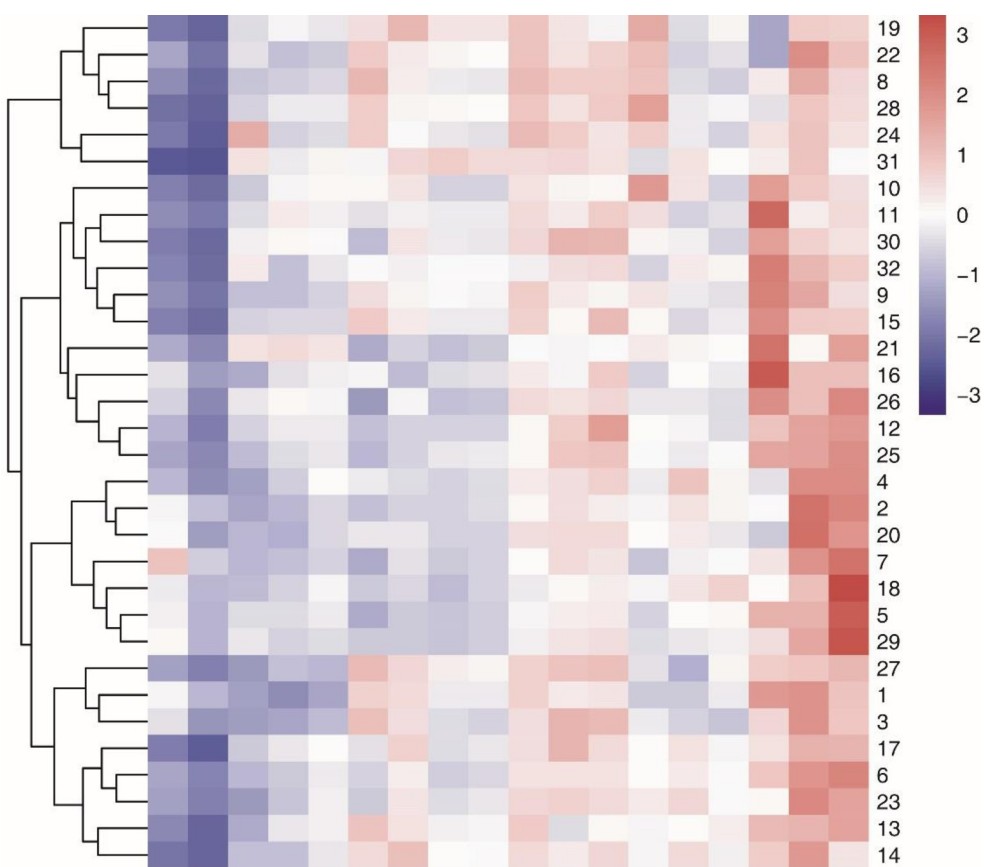

**Fig 1. System clustering of tolerance to low Fe stress in 32 maize hybrids.** The number represents the serial number of cultivars in S1 Table.

were higher (low Fe tolerance coefficient > 1, S2 Table). System clustering analysis of the above 18 response variables showed that the 32 tested maize hybrids were clustered into three types (Fig 1). Among them, 6 cultivars (numbers 19, 22, 8, 28, 24, and 31, which included ZH 2 and Zhengda 619) were resistant to low Fe stress and are Fe-efficient maize cultivars; 11 cultivars (10, 11, 30, 32, 9, 15, 21, 16, 26, 12, and 25, which included Zhongdan 808 and Xianyu 508) were resistant to low Fe stress and are Fe-neutral maize cultivars; 15 cultivars (4, 2, 20, 7, 18, 5, 29, 27, 1, 3, 17, 6, 23, 13, and 14, which included CD 418 and hualongyu 8) were sensitive to low Fe stress and are Fe-inefficient maize cultivars. The low Fe tolerance coefficients decreased for 15 indices in the low Fe treatment, with mean values for the three types of cultivars, Fe-efficient, Fe-neutral and Fe-inefficient, of 0.78, 0.71, and 0.66, respectively. The mean indices for the cultivars, which increased their Fe tolerance coefficients under the low Fe treatment, were 0.90, 1.07, and 1.15, respectively. Under the low Fe treatment, Fe-efficient maize cultivars could effectively control variation in each index and better maintain normal growth.

## Morphological characteristics

In the low Fe treatment, plant height and stem diameter of maize seedlings decreased significantly. Greater decreases were observed when plants were stressed for a longer period (Fig 2). The plant height and stem diameter of these two maize cultivars with different Fe efficiency responses to low Fe stress were significantly different. The decreases in plant height and stem

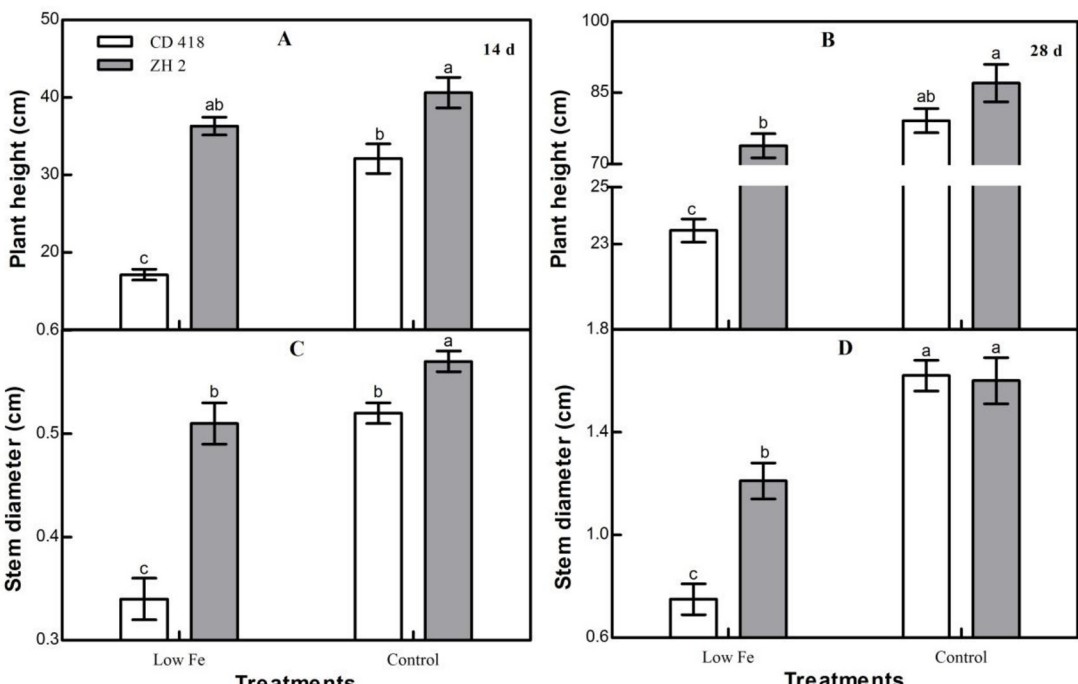

**Fig 2. Plant heights and stem diameters of two maize cultivars grown under different Fe concentrations for 14 or 28 d.**
Data are the mean ± SE of three replicate pots. Values with different lowercase letters are significantly different at $p < 0.05$.

diameter of the Fe-efficient cultivar ZH 2 were notably lesser than those of the Fe-inefficient cultivar CD 418. The differences in plant height and stem diameters between ZH 2 and CD 418 increased with longer stress times. The decreases in plant height and stem diameter of CD 418 were greater than those of ZH 2 by 30.06% and 24.09%, respectively, 14 days after treatment, and 55.12% and 28.32%, respectively, 28 days after treatment.

Under low Fe stress, the shoot, root, and total dry weight of maize seedlings decreased significantly, while the root to shoot ratio increased markedly and the variation increased at the longer stress time (Table 1). At 14 d after treatment, the shoot dry weight of ZH 2 and CD 418

**Table 1. Dry matter accumulation of maize seedlings grown under different Fe concentrations.**

| Cultivars | Fe treatments | Shoot dry weight (g plant⁻¹) | | Root dry weight (g plant⁻¹) | | Total dry weight (g plant⁻¹) | | Root-to-shoot ratio | |
|---|---|---|---|---|---|---|---|---|---|
| | | 14 d | 28 d | 14 d | 28 d | 14 d | 28 d | 14 d | 28 d |
| CD 418 | Control | 0.34 b | 1.96 b | 0.17 b | 0.71 b | 0.51 b | 2.67 b | 0.50 a | 0.36 b |
| | Low Fe | 0.10 c | 0.22 d | 0.05 c | 0.13 c | 0.15 c | 0.35 d | 0.55 a | 0.61 a |
| ZH 2 | Control | 0.49 a | 3.55 a | 0.26 a | 1.13 a | 0.74 a | 4.67 a | 0.52 a | 0.32 b |
| | Low Fe | 0.37 b | 1.63 c | 0.12 b | 0.51 b | 0.49 b | 2.14 c | 0.33 b | 0.32 b |
| *F* value | Cultivar (C) | 80.18** | 299.56** | 18.05** | 28.72** | 233.49** | 317.67** | 27.62** | 69.71** |
| | Iron (Fe) | 57.08** | 548.41** | 42.89** | 63.72** | 255.45** | 519.62** | 14.85** | 38.44** |
| | C×Fe | 6.34** | 1.23ns | 0.13ns | 0.11ns | 8.92** | 1.07ns | 39.78** | 38.44** |

Mean values with different lowercase letters are significantly different at $p < 0.05$; within cultivars, values with different uppercase letters are significantly different at
$p < 0.05$ according to the least significant difference test.

** $p < 0.01$; * $p < 0.05$

ns, not significant.

**Table 2. Fe accumulation of maize seedlings grown under different Fe concentrations.**

| Cultivars | Fe treatments | Shoot Fe accumulation (ug plant$^{-1}$) | | Root Fe accumulation (ug plant$^{-1}$) | | Total Fe accumulation (ug plant$^{-1}$) | | Root-to-shoot Fe ratio | |
|---|---|---|---|---|---|---|---|---|---|
| | | 14 d | 28 d | 14 d | 28 d | 14 d | 28 d | 14 d | 28 d |
| CD 418 | Control | 116.35 b | 318.46 b | 322.10 b | 994.47 b | 438.45 b | 1312.93 b | 2.76 b | 3.11 |
| | Low Fe | 26.67 d | 32.04 d | 20.76 c | 58.39 c | 47.44 d | 90.43 d | 0.78 c | 1.82 |
| ZH 2 | Control | 184.65 a | 622.15 a | 587.27 a | 1262.72 a | 771.92 a | 1884.88 a | 3.18 a | 2.03 |
| | Low Fe | 57.68 c | 247.87 c | 48.41 c | 185.28 c | 106.09 c | 433.15 c | 0.84 c | 0.76 |
| *F* value | Cultivar (C) | 363.70** | 251.50** | 44.34** | 12.03* | 86.44** | 57.65** | 4.64$^{ns}$ | 109.44** |
| | Iron (Fe) | 730.62** | 406.74** | 365.08** | 312.48** | 627.86** | 492.83** | 368.71** | 154.83** |
| | C×Fe | 51.29** | 7.19* | 29.18** | 1.54$^{ns}$ | 42.46** | 3.62$^{ns}$ | 2.43$^{ns}$ | 0.01$^{ns}$ |

Mean values with different lowercase letters are significantly different at $p < 0.05$; within cultivars, values with different uppercase letters are significantly different at $p < 0.05$ according to the least significant difference test.

** $p < 0.01$

* $p < 0.05$

ns, not significant.

decreased by 23.92% and 71.48%, the root dry weight decreased by 51.74% and 68.63%, and the total dry weight decreased by 33.48% and 70.52%, respectively, whereas the root to shoot ratio decreased by 36.87% and increased by 9.97%, respectively. At 28 d after treatment, the shoot dry weight of ZH 2 and CD 418 decreased by 54.04% and 89.00%, respectively, the root dry weight decreased by 55.17% and 81.42%, and the total dry weight decreased by 54.31% and 86.99%, whereas the root to shoot ratio increased by 0.12% and 69.41%, respectively. Under low Fe stress, the decreases in shoot, root, and total dry weight, and the increases in the Fe-efficient cultivar ZH 2 were significantly lower than those of the Fe-inefficient cultivar CD 418. In addition, the differences in dry weight between the two cultivars increased with longer stress times.

The shoot, root, and total Fe accumulation and the root to shoot Fe ratio were significantly lower in the low Fe treatment (Table 2). At 14 days after treatment, the shoot, root, total Fe accumulation, and root to shoot Fe ratio of ZH 2 decreased by 68.76%, 91.76%, 86.26%, and 73.43%, respectively, and those of CD 418 decreased by 77.08%, 93.55%, 89.18%, and 71.75%, respectively. At 28 days after treatment, the shoot, root, total Fe accumulation, and root to shoot Fe ratio of ZH 2 decreased by 60.16%, 85.33%, 77.02%, and 62.29%, respectively, and those of CD 418 decreased by 89.94%, 94.13%, 93.11%, and 41.39%, respectively. The decreases in shoot, root, and total Fe accumulation of the Fe-efficient cultivar ZH 2 were notably lower than those of the Fe-inefficient cultivar CD 418, whereas the decreases in root to shoot Fe ratios were considerably higher than those of CD 418. Under low Fe stress, the Fe-efficient cultivar ZH 2 not only absorbed Fe more efficiently than the Fe-inefficient cultivar CD 418, but also promoted the transportation of Fe from root to shoot to maintain the shoot growth.

Low Fe stress had significant effects on maize seedling Fe physiological efficiency (Fig 3). Under low Fe stress, Fe physiological efficiencies of both maize cultivars were significantly higher than under control conditions (Fig 3). The two maize cultivars with different Fe efficiencies responded with differential changes in Fe physiological efficiency under low Fe stress. At 14 and 28 days after treatment, the Fe physiological efficiency of ZH 2 increased by 383.41% and 96.87%, respectively, while Fe physiological efficiency of CD 418 increased by only 170.92% and 88.75%, respectively. Under low Fe stress, the Fe-efficient cultivar ZH 2 had higher Fe physiological efficiency than the Fe-inefficient cultivar CD 418, suggesting that ZH 2 has greater adaptability to low Fe environments than CD 418.

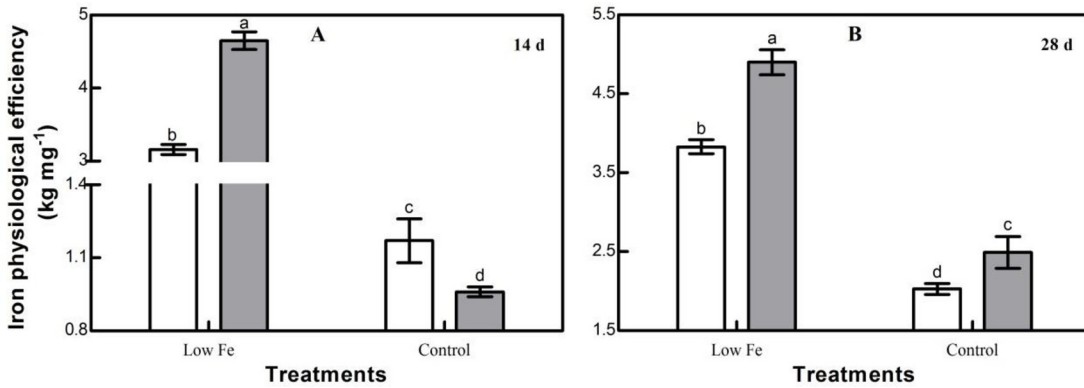

**Fig 3. Fe physiological efficiencies of two maize cultivars grown under different Fe concentrations.** Data are the mean ± SE of three replicate pots. Values with different lowercase letters are significantly different at $p < 0.05$.

The root volume, root length, number of root tips, and root surface area of maize seedlings were significantly lower in the low Fe treatment compared with the control, whereas the average root diameter was increased and there were obvious differences between cultivars (Table 3). Under low Fe stress, the root volume, root length, number of root tips, and root surface area of ZH 2 decreased by 24.73%, 25.16%, 33.96%, and 25.42%, respectively, and the average root diameter increased by 1.30%, while those of CD 418 changed by 28.09%, 50.29%, 59.41%, 37.71%, and 25.13%, respectively. Variation in root volume, root length, number of root tips, root surface area, and average root diameter of the Fe-efficient cultivar ZH 2 was significantly lower than in the Fe-inefficient cultivar CD 418. The root volume, root length, number of root tips, and root surface area of ZH 2 were significantly higher than those of CD 418 under both control and low Fe treatments. They were higher by 47.66%, 47.09%, 27.65%, and 53.66% respectively under the control treatment, and higher by 54.55%, 121.46%, 107.67%, and 83.97%, respectively under the low Fe treatment. The differences in root morphological characteristics between the two cultivars in the low Fe treatment were significantly higher than those under the control treatment. Compared with CD 418, ZH 2 maintained higher root growth under low Fe stress than CD 418, thereby improving its adaptability to a low Fe environment.

**Table 3. Root morphological characteristics of maize seedlings.**

| Cultivars | Fe treatments | Root volume (cm³) | Root length (cm) | Average root diameter (mm) | Number of root tips | Root surface area (cm²) |
|---|---|---|---|---|---|---|
| CD 418 | Control | 4.32 b | 1442.07 b | 0.59 b | 3609.67 b | 268.21 b |
| | Low Fe | 3.10 c | 716.88 c | 0.74 a | 1465.33 c | 167.07 c |
| ZH 2 | Control | 6.37 a | 2121.19 a | 0.62 b | 4607.67 a | 412.12 a |
| | Low Fe | 4.80 b | 1587.58 b | 0.63 b | 3043.00 b | 307.35 b |
| F value | Cultivar (C) | 61.41** | 48.62** | 2.03ns | 15.42** | 92.87** |
| | Iron (Fe) | 33.95** | 32.07** | 6.14* | 31.98** | 48.75** |
| | C×Fe | 0.58ns | 0.74ns | 4.95ns | 0.78ns | 0.02ns |

Mean values with different lowercase letters are significantly different at $p < 0.05$; within cultivars, values with different uppercase letters are significantly different at $p < 0.05$ according to the least significant difference test.

** $p < 0.01$

* $p < 0.05$

ns, not significant.

Under low Fe stress, the root activity of maize seedlings decreased significantly, indicating that low Fe stress inhibited root activity and reduced the root Fe absorption ability (Fig 4A). The root activity of ZH 2 was significantly higher than CD 418 under the control and low Fe treatments by 7.13% and 36.07%, respectively. Under low Fe stress, ZH 2 maintained higher root activity than CD 418, which increased absorption of Fe. Under low Fe stress, the Fe-dissolution ability per mass of roots in the maize seedlings increased significantly, while the Fe-dissolution ability per plant decreased significantly (Fig 4B and 4C). Compared with the control treatment, in the low Fe treatment the Fe-dissolution ability per mass of roots in ZH 2 and CD 418 increased by 62.05% and 14.96%, respectively, and the Fe-dissolution ability per plant decreased by 21.76% and 64.04%, respectively. The increase in Fe-dissolution ability per mass of roots in ZH 2 was significantly higher than that of CD 418, whereas the decrease in Fe-dissolution ability per plant of ZH 2 was significantly lower than that of CD 418. In the low Fe treatment, the Fe-dissolution ability per mass of roots and per plant in ZH 2 were significantly higher in CD 418 by 18.26% and 173.51%, respectively. Overall, the $Fe^{3+}$ chelating ability of the Fe-efficient cultivar ZH 2 was notably higher than that of the Fe-inefficient cultivar CD 418.

## Photosynthetic characteristics

Under low Fe stress, leaf area, relative chlorophyll content, and the net photosynthetic rate of maize seedlings were significantly reduced, but significant differences were observed between cultivars (Fig 5). Compared with the control treatment, the leaf area, relative chlorophyll content, and net photosynthetic rate of ZH 2 decreased by 19.76%, 10.15%, and 30.51%, respectively, under low Fe stress, whereas those of CD 418 decreased by 77.28%, 56.03%, and 79.75%,

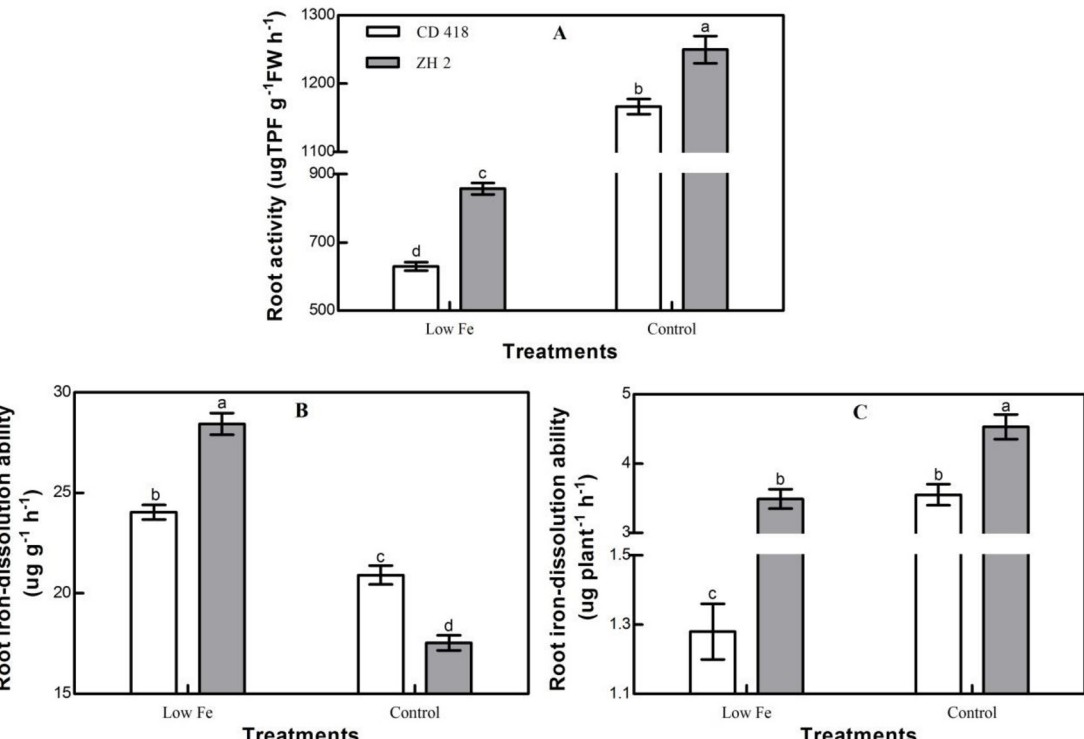

**Fig 4. Root activity and Fe-dissolution ability of two maize cultivars grown under different Fe concentrations.** Data are the mean ± SE of three replicate pots. Values with different lowercase letters are significantly different at $p < 0.05$.

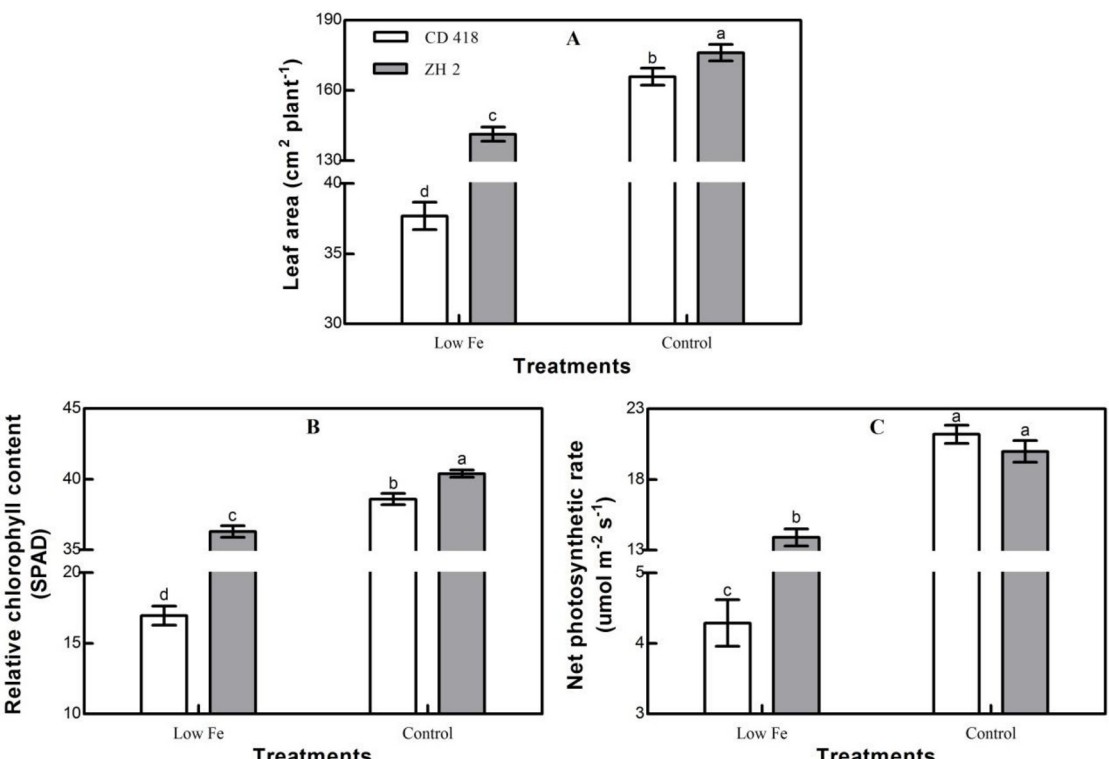

**Fig 5. Photosynthetic characteristics (leaf area, relative chlorophyll content, and net photosynthetic rate) of two maize cultivars grown under different Fe concentrations.** Data are the mean ± SE of three replicate pots. Values with different lowercase letters are significantly different at p < 0.05.

respectively. The decreases in leaf area, relative chlorophyll content, and net photosynthetic rate of CD 418 were significantly higher than those of ZH 2. Under low Fe stress, the Fe-efficient cultivar ZH 2 maintained a higher photosynthetic area and photosynthetic rate than the Fe-inefficient cultivar CD 418. This would ensure the supply of photosynthetic products and improve the adaptability of ZH 2 to low Fe environments.

## Physiological characteristics

Compared with the control treatment, the soluble protein content in maize seedling leaves in the low Fe treatment increased significantly, and there were significant differences between cultivars (Fig 6A). The soluble protein content of ZH 2 was significantly higher than CD 418 under control and low Fe treatments by 10.84% and 11.62%, respectively. The Fe-efficient cultivar ZH 2 had higher protein synthesis ability than the Fe-inefficient cultivar CD 418, and hence, greater adaptability to low Fe stress.

Compared with the control, under low Fe stress, the POD activity of ZH 2 increased by 5.52%, and that of CD 418 decreased by 5.96%; the CAT activity of both cultivars decreased by 49.32% and 43.88%, respectively (Fig 6B and 6C). The activities of POD and CAT in ZH 2 were significantly higher than those in CD 418 under control and low Fe treatments; POD activities were higher by 7.95% and 21.13%, respectively, and CAT activities were higher by 53.73% and 38.83%, respectively. These results suggest that the Fe-efficient cultivar ZH 2 had higher active oxygen scavenging ability than the Fe-inefficient cultivar CD 418, allowing it to better adapt to a low Fe environment than CD 418.

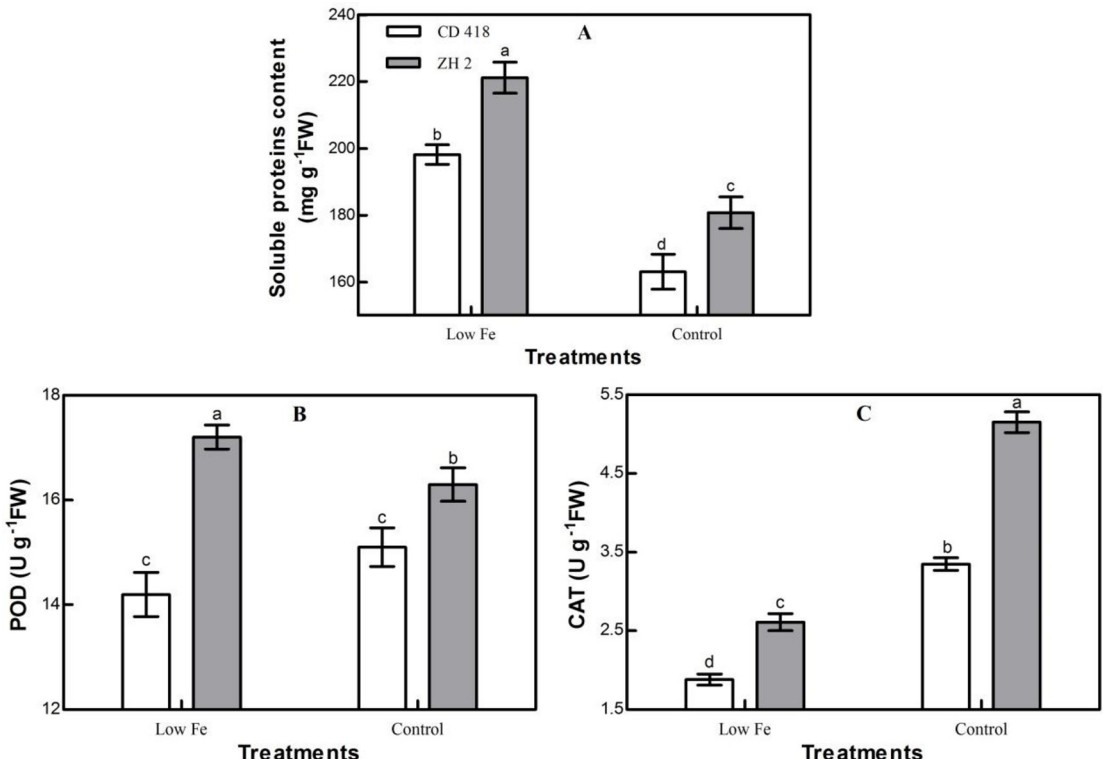

**Fig 6. Physiological characteristics (soluble protein content, POD, and CAT) of two maize cultivars grown under different Fe concentrations.** FW: fresh weight. Data are the mean ± SE of three replicate pots. Values with different lowercase letters are significantly different at $p < 0.05$.

## Discussion

### Cultivar differences in root Fe uptake ability

Under low Fe stress, plant root systems adapt through a series of morphological and physiological changes [14, 25]. Root morphology of maize seedlings is directly related to Fe efficiency under low Fe stress, and good root morphology and physiological activity are important characteristics of efficient Fe utilization [26]. In addition, Fe-deficient conditions increase the reduction potential of $Fe^{3+}$, and acidity of the rhizosphere, but reduce Fe content and Fe accumulation in wheat plants [27]. In agreement with these previous results [28], we found that under low Fe stress, the root volume, root length, number of root tips, root surface area, and root dry weight of maize seedlings decreased significantly, while the root to shoot ratio and average root diameter increased significantly. In addition, the root to shoot ratio increased significantly, indicating that in conditions of limited plant matter accumulation, the photosynthetic products synthesized in shoots were preferentially distributed to the root system to maintain root system growth. The increase in the root to shoot ratio was significantly higher in CD 418 than in ZH 2, indicating that the Fe-inefficient cultivar CD 418 was more sensitive to low Fe stress than the Fe-efficient cultivar ZH 2. This finding is inconsistent with results of Xu et al. [13], who reported that the root to shoot ratio was higher in the Fe-efficient cultivar Ye478 than in the Fe-inefficient cultivar Wu312. The main reason for this discrepancy is that the Fe-efficient cultivar ZH 2 selected for our study has a different Fe-efficiency mechanism than that of Ye478; ZH 2 was efficient at Fe absorption, whereas Ye478 has efficient root to shoot transport and redistribution of Fe in the plant [13].

To obtain sufficient nutrients, plants can grow their root system (as measured by root length, root volume, root dry weight, and root surface area) or increase the ability of the roots to absorb nutrients [29]. Dasgan et al. [30] showed that Fe-efficient tomato plants have more lateral roots than Fe-inefficient tomatoes grown under low Fe conditions; the increase in lateral roots increases the absorption surface area of tht roots. Shi et al. [31] studied the Fe nutrition efficiency of six wheat genotypes and found that differences in Fe absorption were mainly caused by differences in root surface area. We found that the root volume, root length, number of root tips, root surface area, and root dry weight of ZH 2 were significantly higher than those of CD 418 under both control and low Fe treatments. The differences in root characteristics between the two cultivars under the low Fe treatment were higher than under the control treatment, indicating that the ZH 2 plants had a more developed root system. This resulted in significantly higher Fe accumulation in ZH 2 than in CD 418, especially under Fe deficiency conditions. As a mechanism II plant, the root Fe-dissolution ability (the amount of mugineic acid secretion) of maize directly affects its Fe absorption ability [15]. Our results showed that the Fe-dissolution ability per mass of roots increased significantly, while the Fe-dissolution ability per plant decreased significantly under low Fe stress, consistent with results of Yin et al. [27] in wheat. The increase in Fe-dissolution ability per mass of roots was significantly higher in ZH 2 than in CD 418, resulting in a significantly higher Fe dissolution rate per mass of roots in ZH 2 than in CD 418. Thus, under low Fe stress conditions the Fe absorption capacity per mass of roots in the Fe-efficient cultivar ZH 2 was higher than in the Fe-inefficient CD 418. Overall, efficient Fe absorption in the Fe-efficient cultivar ZH 2 compared with the Fe-inefficient cultivar CD 418 was due to a more developed root system and a higher Fe absorption capacity of roots.

## Adaptation mechanism of maize seedlings to low Fe stress

Photosynthesis is the basis of plant growth. Fe deficiency inhibits photosynthesis by harming chloroplast structure and reducing chlorophyll synthesis [15]. Hence, under low Fe stress the net photosynthetic rate, stomatal conductance, and transpiration rate of pea leaves decreases, and the intercellular $CO_2$ concentration increases [32]. Consistent with previous results [32], we found that the effective photosynthetic area, chlorophyll content, and net photosynthetic rate of maize decreased significantly under low Fe stress. However, the decreases in photosynthetic parameters were significantly lower in ZH 2 compared with CD 418, as observed by the overall higher effective photosynthetic area and photosynthetic rate of ZH 2 than those of CD 418. Due to an adequate supply of photosynthetic products, the plant height, stem diameter, shoot dry weight, root dry weight, and total dry weight in ZH 2 plants were significantly higher than CD 418 plants under low Fe stress, and the root to shoot ratio was significantly lower. These results indicate that the Fe-efficient cultivar ZH 2 could effectively control the decline of photosynthetic area and photosynthetic rate under low Fe stress, maintain higher matter accumulation, and coordinate the growth of shoot and root, hence improving its overall adaptability to a low Fe environment.

When plants are stressed, they can activate defense mechanisms to adapt to the adverse environment [25]. For example, the enzymes POD and CAT can eliminate accumulation of reactive oxygen species in plants, keep free radicals in plant cells at a low level, reduce membrane damage caused by membrane lipid peroxidation, and maintain normal plant life activities. In addition, increase in organic substance content, such as soluble protein, can improve the osmotic adjustment ability of leaves, delay leaf senescence, and provide carbon and nitrogen sources for the synthesis of plant organic matter [22, 23]. CAT and POD activities in maize seedlings decreased under low Fe stress, which led to the decrease of active oxygen

scavenging ability and the impairment of membrane system function. Similarly, soluble protein, osmoregulation substance, increased significantly under low Fe stress, which facilitated the balance of osmotic potential in maize-seedling cells to a certain degree. Soluble protein content, POD activity, and CAT activity were significantly higher in ZH 2 than in CD 418 under control as well as low Fe treatments. Under low Fe stress, active oxygen scavenging activity decreased, and the internal membrane system was damaged, but the compensation through increased content of osmotic regulators such as soluble protein, provided some degree of osmotic potential balance. ZH 2 had higher soluble protein content, CAT activity, and POD activity than CD 418, indicating that the Fe-efficient cultivar ZH 2 had better osmotic regulation ability and active oxygen scavenging ability than the Fe-inefficient cultivar CD 418. This is one of the important physiological mechanisms of Fe-efficient cultivars that can be used to adapt to low Fe stress.

## Conclusion

Selection of Fe-efficient maize cultivars is the simplest and most effective way to alleviate low Fe stress and improve maize yield in Fe-deficient soils. In our study, six Fe-efficient cultivars, including ZH 2, and 15 Fe-inefficient cultivars, such as CD 418, were selected from 32 major maize hybrids in Southwest China. We found that low Fe stress resulted in significant decreases in root volume, root length, the number of root tips, root surface area, and root dry weight in both cultivars, and increased root to shoot ratio, average root diameter, and Fe-dissolution ability per mass of roots. However, the increase of Fe-dissolution ability per mass of roots in ZH 2 was higher than that of CD 418, while for other parameters, the low Fe stress-induced changes in ZH 2 were less pronounced than in CD 418. Moreover, under low Fe stress, the abovementioned growth factors were significantly higher in ZH 2 than in CD 418. Therefore, compared with the Fe-inefficient cultivar CD 418, the Fe-efficient cultivar ZH 2 had a more developed root system and stronger Fe absorption capacity per mass of roots under low iron stress, promoted the efficient absorption of Fe, maintained a higher photosynthetic area and photosynthetic rate, thereby, ensuring accumulation of photosynthetic products. Further, it had higher soluble protein content and activities of catalase and peroxidase that permitted strong osmotic regulation and scavenging ability, which is an important physiological mechanism for ZH 2 adaptation to low Fe stress.

## Supporting information

**S1 Data.**
(XLSX)

**S1 Table. List of maize cultivars.**
(DOCX)

**S2 Table. Differences in the low Fe tolerance coefficient of maize hybrids.**
(DOCX)

## Author Contributions

**Data curation:** Wengjing Long.

**Formal analysis:** Qiang Li.

**Funding acquisition:** Jichao Yuan.

**Investigation:** Nianxin Wan.

**Methodology:** Dongju Feng.

**Software:** Yong Zhou.

**Writing – review & editing:** Fanlei Kong.

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
