## [Decision Letter · Decision Letter 0]

21 May 2020

PONE-D-20-12291

Morphological and physiological characteristics of maize seedling adapted to low iron stress

PLOS ONE

Dear Dr. Jichao,

Thank you for submitting your manuscript to PLOS ONE. After careful consideration, we feel that it has merit but does not fully meet PLOS ONE’s publication criteria as it currently stands. Therefore, we invite you to submit a revised version of the manuscript that addresses the points raised during the review process.

Spellings and English language needs to be checked thoroughly. Introduction, main claims of the paper are not properly placed in the context of previous literature. In results, drafting of many sentences need to be improved. In discussion, there is a lack of mechanistic approach at some points.

We look forward to receiving your revised manuscript.

Kind regards,

Basharat Ali, Ph.D

Academic Editor

PLOS ONE

Journal Requirements:

Reviewers' comments:

Reviewer's Responses to Questions

**Comments to the Author**

1. Is the manuscript technically sound, and do the data support the conclusions?

Reviewer #1: Yes

Reviewer #2: Partly

2. Has the statistical analysis been performed appropriately and rigorously? 

Reviewer #1: Yes

Reviewer #2: Yes

3. Have the authors made all data underlying the findings in their manuscript fully available?

Reviewer #1: Yes

Reviewer #2: No

4. Is the manuscript presented in an intelligible fashion and written in standard English?

Reviewer #1: Yes

Reviewer #2: No

5. Review Comments to the Author

Reviewer #1: Dear editor,

Regarding this manuscript, the manuscript is well written and after some minor revision could be suitable for publication. The authors need to significantly improve its quality. There is significant work to be done on this manuscript before acceptance, particularly in materials and methods, discussion and conclusion for more information see the attached file.

Regards

HM Rawashdeh

General comments

This is an interesting article. It addresses important issues of iron deficiency, especially in arid and semi-arid calcareous soil regions and the authors have done a good job. However, I have a few comments and questions and I hope this can contribute to a better and clearer manuscript.

Language: The present language quality is not good enough and needs to be improved. The manuscript would benefit from a thorough proofread by a native English speaker to improve grammar.

Abstract: I would prefer a results section with fewer details and numbers. The most important findings can be addressed, but I would not include all findings. The abstract is not well balanced.

Introduction: It is a good introduction. It contextualizes and gives important background knowledge.

Materials and methods: it is good but the authors need to give more details about their experiments such as the number of replications, experimental design, and statistical analysis that they used in their experiments.

Results: Overall it is a good result and clear.

Discussion: Overall it is a good discussion; I think the discussion would be even better if the authors use more appropriate references to have a good discussion.

Conclusion: the conclusion is too long; the authors can be reduced without losing the key.

References: some of the references are old [17] and [18]. All the references should be presented according to journal guidelines.

Reviewer #2: Title: Morphological and physiological characteristics of maize seedling adapted to low iron stress

The ms by Long et al., mainly reports the morphological and physiological responses of two rice cultivars to iron deficiency in terms of root morphology, growth, photosynthesis, and antioxidant enzymes. Although the investigation is interesting and comprehensive, but need a through revision prior to publications. There are numerous flaws in different sections of the ms, as indicated below in the specific comments.

Abstract: Clearly add the treatments’ description. Levels of Fe supply? If possible, add the numeric description of some key parameters. Conclude the abstract section (Key findings in a single sentence)

Better to change the title, reporting key findings of your study.

Introduction: Introduction section contains various unnecessary statements. Remain focused on the topic. Begin with the broadest scope and get progressively narrower.

Add a clear-cut research hypothesis.

Whether the Fe deficiency is a problem in southwest China. Add such information in introduction. I think, its not a problem in soils with pH<7.

Objectives should be specific and clear. Morphological and physiological parameters were only investigated for two cultivars (not for all).

Methods:

It is necessary to clearly substantiate the use of two cultivars (Exp 2), since there are no explanations in the text of the article. It would nice if authors shortly indicate some major traits of chosen maize cultivars. Why these cultivars were selected to experiment? Based of which parameters?

L113: The seeds were then germinated (How many?) Similarly, how many seedlings were transferred to pots.

Add the details of software used.

If possible, add the brief methodology for the measurement of different parameters. How about sampling method/time/amount?

Clearly specify the details on statistical analysis separately for exp 1 and 2. It can not be same.

Information regarding experimental design is missing. Be consistent regarding the use of abbreviations…

Results. Despite the large amount of results obtained by the authors, they look like a set of data that are not interconnected.

Language needs substantial improvement. There are several grammatical and typo mistakes. Many statements are confusing/unclear. e.g. L203: the decreases in plant height and stem diameter (compared with what?)

Why the authors only measured only CAT and POD, if the data regarding other oxidative stress indicators (e.g ROS) is not present.

Discussion should be merely based on the observed findings Answer the question posed in introduction, and correlate your finding with the existing knowledge. Avoid the repetition. i.e description of results. Check whether the format of all references is according to the journal format.

Conclusion: Just report the key findings, it should not be detailed summary.

6. PLOS authors have the option to publish the peer review history of their article (what does this mean?). If published, this will include your full peer review and any attached files.

Reviewer #1: Yes: Hamzeh Mohmmad Rawashdeh, National Agricultural Research Center (NARC), Amman, Jordan

Reviewer #2: Yes: Saddam Hussain

---

## [Author Response · Author response to Decision Letter 0]

10 Jul 2020

Dear Editors and Reviewers:

Thank you for your letter and for the reviewers’ comments concerning our manuscript entitled “Morphological and physiological characteristics of maize seedling adapted to low iron stress” (PONE-D-20-12291). Those comments are all valuable and very helpful for revising and improving our manuscript, as well as the important guiding significance to our researches. We have studied comments carefully and have made correction which we hope meet with approval. Revised portion are marked in red in the paper. The main corrections in the paper and the responds to the reviewer’s comments are as following: 

Responds to the reviewer’s comments:

Reviewer: 1

Language: The present language quality is not good enough and needs to be improved. The manuscript would benefit from a thorough proofread by a native English speaker to improve grammar.

Response to comment: Our revised manuscript has been polished and revised by senior editor of Editage.

Abstract: I would prefer a results section with fewer details and numbers. The most important findings can be addressed, but I would not include all findings. The abstract is not well balanced.

Response to comment: We have revised the abstract to present some details and numbers.

Introduction: It is a good introduction. It contextualizes and gives important background knowledge.

Response to comment: There is no need to answer.

Materials and methods: it is good but the authors need to give more details about their experiments such as the number of replications, experimental design, and statistical analysis that they used in their experiments.

Response to comment: We added the number of replications, experimental design, and statistical analysis that they used in their experiments in the revised manuscript.

Results: Overall it is a good result and clear.

Response to comment: There is no need to answer.

Discussion: Overall it is a good discussion; I think the discussion would be even better if the authors use more appropriate references to have a good discussion.

Response to comment: We used some more appropriate references to have a good discussion in the revised manuscript.

Conclusion: the conclusion is too long; the authors can be reduced without losing the key.

Response to comment: We reduced the conclusion without losing the key. 

References: some of the references are old [17] and [18]. All the references should be presented according to journal guidelines.

Response to comment: We replaced the old [17] and [18] with new references, and presented all the references according to journal guidelines.

Reviewer: 2

Abstract: Clearly add the treatments’ description. Levels of Fe supply? If possible, add the numeric description of some key parameters. Conclude the abstract section (Key findings in a single sentence)

Response to comment: We have revised the abstract, added Fe supply levels and numeric description of some key parameters.

Better to change the title, reporting key findings of your study.

Response to comment: We changed the title to focused on root morphological and physiological characteristics.

Introduction: Introduction section contains various unnecessary statements. Remain focused on the topic. Begin with the broadest scope and get progressively narrower.

Add a clear-cut research hypothesis.

Whether the Fe deficiency is a problem in southwest China. Add such information in introduction. I think, its not a problem in soils with pH<7.

Objectives should be specific and clear. Morphological and physiological parameters were only investigated for two cultivars (not for all).

Response to comment: We have revised the introduction section and added a clear-cut research hypothesis.

Sichuan Province is a big province of maize planting and consumption in China. More than 1/3 of dry land in the region is calcareous purple soil, and the content of available iron in soil is low, which seriously restricts the improvement of maize yield.

 We have added this information in introduction.

Of course, iron deficiency is serious in calcareous soil (pH>7).

We have added the objectives. In this study, we selected two cultivars with the greatest difference in adaptability to low iron stress to study morphological and physiological parameters, mainly considering the workload and timeliness of the experiment. Of course, in the follow-up study, we are also considering the differences of all cultivars to present more systematic research results.

Methods:

It is necessary to clearly substantiate the use of two cultivars (Exp 2), since there are no explanations in the text of the article. It would nice if authors shortly indicate some major traits of chosen maize cultivars. Why these cultivars were selected to experiment? Based of which parameters?

L113: The seeds were then germinated (How many?) Similarly, how many seedlings were transferred to pots.

Add the details of software used.

If possible, add the brief methodology for the measurement of different parameters. How about sampling method/time/amount?

Clearly specify the details on statistical analysis separately for exp 1 and 2. It can not be same.

Information regarding experimental design is missing. Be consistent regarding the use of abbreviations…

Response to comment: The two cultivars are the biggest difference in iron efficiency through cluster analysis and principal component analysis based on the data in the first Experiment. We have added shortly indicate some major traits of chosen maize cultivars in the revised manuscript. 

 We added the number of seedlings transferred to pots in the revised manuscript.

We added the details of software used, and sampling method/time/amount in the revised manuscript.

We clearly specified the details on statistical analysis separately for exp 1 and 2 in the revised manuscript.

We added the experimental design information in the revised manuscript.

Results. Despite the large amount of results obtained by the authors, they look like a set of data that are not interconnected.

Response to comment: We reorganized the results section to make them more interconnected in the revised manuscript.

Language needs substantial improvement. There are several grammatical and typo mistakes. Many statements are confusing/unclear. e.g. L203: the decreases in plant height and stem diameter (compared with what?)

Response to comment: Our revised manuscript has been polished and revised by senior editor of Editage, and rewrite the confusing part in the revised manuscript.

Why the authors only measured only CAT and POD, if the data regarding other oxidative stress indicators (e.g ROS) is not present.

Response to comment: There are many oxidative stress indicators of plant (e.g. CAT, POD, SOD, ROS), we selected CAT and POD to reflect the differences between the two maize cultivars in the study, so not all the oxidative stress indicators were measured. 

Discussion should be merely based on the observed findings Answer the question posed in introduction, and correlate your finding with the existing knowledge. Avoid the repetition. i.e description of results. Check whether the format of all references is according to the journal format.

Response to comment: We revised the discussion and formatted of all references according to the journal format.

Conclusion: Just report the key findings, it should not be detailed summary.

Response to comment: We reduced the conclusion.

---

## [Decision Letter · Decision Letter 1]

22 Jul 2020

PONE-D-20-12291R1

Root morphological and physiological characteristics of maize seedlings adapted to low iron stress

PLOS ONE

Dear Dr. Yuan,

Thank you for submitting your manuscript to PLOS ONE. After careful consideration, we feel that it has merit but does not fully meet PLOS ONE’s publication criteria as it currently stands. Therefore, we invite you to submit a revised version of the manuscript that addresses the points raised during the review process.

ACADEMIC EDITOR: Please see reviewers comments, mostly they are encouraging to enhance the English language, and brief methodology for the measurements of different morphological and physiological attributes is needed. 

We look forward to receiving your revised manuscript.

Kind regards,

Basharat Ali, Ph.D

Academic Editor

PLOS ONE

Reviewers' comments:

Reviewer's Responses to Questions

**Comments to the Author**

1. If the authors have adequately addressed your comments raised in a previous round of review and you feel that this manuscript is now acceptable for publication, you may indicate that here to bypass the “Comments to the Author” section, enter your conflict of interest statement in the “Confidential to Editor” section, and submit your "Accept" recommendation.

Reviewer #1: All comments have been addressed

Reviewer #2: (No Response)

2. Is the manuscript technically sound, and do the data support the conclusions?

Reviewer #1: Yes

Reviewer #2: Yes

3. Has the statistical analysis been performed appropriately and rigorously? 

Reviewer #1: Yes

Reviewer #2: Yes

4. Have the authors made all data underlying the findings in their manuscript fully available?

Reviewer #1: Yes

Reviewer #2: Yes

5. Is the manuscript presented in an intelligible fashion and written in standard English?

Reviewer #1: No

Reviewer #2: Yes

6. Review Comments to the Author

Reviewer #1: English must be enhanced

Even though suggestions in his respect were provided by former reviewers, they are not respected, and the manuscript was not edited as a result. The authors must take into account the above-mentioned suggestions and make the suitable corrections.

Reviewer #2: Authors have addressed all the comments raised by me, and the revised draft is much improved. However, there are still few minor issues, which need to be addressed prior to publication.

-Be consistent regarding abbreviations. All the abbreviations should be well defined at first mentioned place. Line36, CAT and POD are unexplained.

-Keywords are not sufficient, and do not cover the main theme of ms.

-In methods, add appropriate reference regarding the selection of treatments, and usage of nutrient solution.

-Add brief methodology for the measurements of different morphological and physiological attributes.

-In statistical analysis, probability level for mean comparison test?

In discussion: Discuss the activities of antioxidant enzymes in relation to Fe supply.

In Figure 6 captions: define 1 U for CAT and POD,

7. PLOS authors have the option to publish the peer review history of their article (what does this mean?). If published, this will include your full peer review and any attached files.

Reviewer #1: No

Reviewer #2: **Yes: **Saddam Hussain

---

## [Author Response · Author response to Decision Letter 1]

16 Aug 2020

Dear Editors and Reviewers:

Thank you for your letter and for the reviewers’ comments concerning our manuscript entitled “Root morphological and physiological characteristics of maize seedlings adapted to low iron stress” (PONE-D-20-12291R1). Those comments are all valuable and very helpful for revising and improving our manuscript, as well as the important guiding significance to our researches. We have studied comments carefully and have made correction which we hope meet with approval. Revised portion are marked in red in the paper. The main corrections in the paper and the responds to the reviewer’s comments are as following: 

Responds to the reviewer’s comments:

Reviewer: 1

English must be enhanced

Even though suggestions in his respect were provided by former reviewers, they are not respected, and the manuscript was not edited as a result. The authors must take into account the above-mentioned suggestions and make the suitable corrections.

Response to comment: Although in the first revision, we have already revised the English of the full manuscript through Editage, which has not achieved good results, we are deeply sorry. After checking the reviewers' comments, we applied for Editage to revise the full manuscript English of the article, hoping to meet the requirements of publication. 

Reviewer: 2

Be consistent regarding abbreviations. All the abbreviations should be well defined at first mentioned place. Line36, CAT and POD are unexplained.

Response to comment: We have revised it in the revised manuscript.

Keywords are not sufficient, and do not cover the main theme of ms.

Response to comment: We changed the keywords “Iron, Chlorophyll, mugineic acid, Zea mays” to “Iron, root morphology, physiological characteristics, Zea mays L.”.

Add brief methodology for the measurements of different morphological and physiological attributes.

Response to comment: In the part of “Sampling and measurements”, we have supplemented the brief methodology for the measurements of different morphological and physiological attributes.

In statistical analysis, probability level for mean comparison test?

Response to comment: Means were compared using least significant difference (LSD) test at the 0.05 level to determine the diﬀerences between the means of each treatment. It has been added in the revised manuscript.

In discussion: Discuss the activities of antioxidant enzymes in relation to Fe supply.

Response to comment: We have supplemented the discussion in the revised manuscript.

In Figure 6 captions: define 1 U for CAT and POD

Response to comment: We have defined 1 U for CAT and POD in the revised manuscript.

---

## [Decision Letter · Decision Letter 2]

31 Aug 2020

Root morphological and physiological characteristics of maize seedlings adapted to low iron stress

PONE-D-20-12291R2

Dear Dr. Jichao Yuan,

We’re pleased to inform you that your manuscript has been judged scientifically suitable for publication and will be formally accepted for publication once it meets all outstanding technical requirements.

Kind regards,

Basharat Ali, Ph.D

Academic Editor

PLOS ONE

Additional Editor Comments (optional):

Reviewers' comments:

Reviewer's Responses to Questions

**Comments to the Author**

1. If the authors have adequately addressed your comments raised in a previous round of review and you feel that this manuscript is now acceptable for publication, you may indicate that here to bypass the “Comments to the Author” section, enter your conflict of interest statement in the “Confidential to Editor” section, and submit your "Accept" recommendation.

Reviewer #1: All comments have been addressed

Reviewer #2: All comments have been addressed

2. Is the manuscript technically sound, and do the data support the conclusions?

Reviewer #1: Yes

Reviewer #2: Yes

3. Has the statistical analysis been performed appropriately and rigorously? 

Reviewer #1: Yes

Reviewer #2: Yes

4. Have the authors made all data underlying the findings in their manuscript fully available?

Reviewer #1: Yes

Reviewer #2: Yes

5. Is the manuscript presented in an intelligible fashion and written in standard English?

Reviewer #1: Yes

Reviewer #2: No

6. Review Comments to the Author

Reviewer #1: The authors have addressed all the comments advanced and constructive suggestions, which helped to improve the quality of their manuscript. I would like to stress out that I support the potential

publication of this paper due to its scientific interest.

Reviewer #2: Authors have improved the ms as per suggestions, therefore, I am happy to recommend acceptance in PLOS ONE.

7. PLOS authors have the option to publish the peer review history of their article (what does this mean?). If published, this will include your full peer review and any attached files.

Reviewer #1: No

Reviewer #2: **Yes: **Saddam Hussain

---

## [Editor Report · Acceptance letter]

4 Sep 2020

PONE-D-20-12291R2 

Root morphological and physiological characteristics in maize seedlings adapted to low iron stress 

Dear Dr. Yuan:

I'm pleased to inform you that your manuscript has been deemed suitable for publication in PLOS ONE. Congratulations! Your manuscript is now with our production department. 

Kind regards, 

on behalf of

Dr. Basharat Ali 

Academic Editor

PLOS ONE